# Research on the Quantitative Method of Cognitive Loading in a Virtual Reality System

**Jian Lv, Xiaoping Xu \* and Ning Ding**

Key Laboratory of Advanced Manufacturing Technology, Ministry of Education, Guizhou University,
Guiyang 550003, China; jlv@gzu.edu.cn (J.L.); dingning19@163.com (N.D.)
\* Correspondence: xiao1648402171@163.com; Tel.: +86-181-8517-0284

**Abstract:** Aimed at the problem of how to objectively obtain the threshold of a user's cognitive load in a virtual reality interactive system, a method for user cognitive load quantification based on an eye movement experiment is proposed. Eye movement data were collected in the virtual reality interaction process by using an eye movement instrument. Taking the number of fixation points, the average fixation duration, the average saccade length, and the number of the first mouse clicking fixation points as the independent variables, and the number of backward-looking times and the value of user cognitive load as the dependent variables, a cognitive load evaluation model was established based on the probabilistic neural network. The model was validated by using eye movement data and subjective cognitive load data. The results show that the absolute error and relative mean square error were 6.52%–16.01% and 6.64%–23.21%, respectively. Therefore, the model is feasible.

**Keywords:** cognitive load; eye movement experiment; virtual reality

## 1. Introduction

A cognitive load is the ratio of task complexity to the cognitive ability required by the user to complete the task, which can be described as the limited capacity of working memory and attention [1]. The cognitive load has a tremendous impact on the user's ability to execute tasks, which is an important humanistic factor directly related to the efficiency of the system operation, job safety, and production efficiency in different fields [2]. In the in-vehicle information system (IVIS), the complex and indiscriminate provision of multiple large sets of data may trigger the cognitive load of drivers, resulting in operational errors and traffic accidents [3]. Therefore, researchers have been conducting quantitative research on the cognitive load, mainly measuring the working memory capacity and selective attention mechanism changes in two stages [1,2,4,5]. Physiological signals (such as heart rate and respiratory rate), brain activity, blood pressure, skin electrical response, pupil diameter, blinking, and gaze are considered biomarkers for quantifying the cognitive load [6,7]. There is an information structure that can effectively quantify the cognitive load in Web browsing and Web shopping, minimize the user's information browsing time, or define the optimal point in time to guide the purchase [8].

Differences in individual cognitive ability and how to enhance the cognitive load affect human cognitive control, which leads to different discoveries of physiological changes as the cognitive load [9], and eye movement technology can objectively measure the cognition of users [10]. The pupil measure is the cognitive activity index (ICA), which assesses the association between expected eye movements and immediate cognitive load [11,12]. The analysis of eye tracking data provides quantitative evidence for the change of the interface layout and its effect on the user's understanding and cognitive load [13]. Many researchers use eye movement behavior data [14–16] to obtain the user's behavior habits and interest difference to judge the user's cognitive load. Among them, Asan et al. [17] studied the physiological index associated with the eye movement tracking technology and cognitive load. These

studies have focused on the use of physiological methods to assess the cognitive load of users but have not yet resolved how to construct a quantitative relationship between physiological indicators and cognitive load.

In addition to analyzing the impact of users' physiological indicators on cognitive load, some researchers have also used machine learning to predict the quantitative cognitive load. The K-NN (k-NearestNeighbor) algorithm has been used to calculate the cognitive load of the user based on, for example, a change in the blood oxygen content of the prefrontal lobe [18,19]. Other studies have shown that both artificial neural networks [20] and classifiers based on linear discriminant analysis [21] can monitor the workload of the EEG (Electroencephalogram) power spectrum in real time. In addition, artificial neural networks [22], aggregation methods [23], and similar approaches have been applied to the data collection of psychophysiological indicators to predict the cognitive load of users.

The main research purpose of this paper was to obtain objective and accurate user cognitive load values in the virtual reality (VR) interactive system. The eye movement test was used, where the number of fixation points of the user was obtained by the eye movement instrument. Additionally, the average fixation duration, average saccade length, the number of fixation points clicked by the first mouse, and the number of backward-looking times were used as the evaluation indexes. A cognitive load evaluation model was then constructed based on the probabilistic neural network, which quantifies the cognitive load and provides a theoretical basis for the design and development of the subsequent virtual reality interactive system.

## 2. Related Work

### 2.1. Multi-Channel Interactive Information Integration in the VR System

To solve the problem that it is difficult to quantify the cognitive load of users in a virtual reality interactive system, in order to reduce the difficulty of interactive cognitive analysis, some researchers have constructed a multi-modal cognitive processing model that integrates touch, hearing, and vision [24]. In order to improve the naturalness and efficiency of interaction, some researchers have also established a multi-modal conceptual model and a system model of human–computer interaction based on the elements of human–computer interaction in command and control [25]. By simulating the process of human brain cognition, this paper studies the interactive behavior of a virtual reality system from cognitive and computational perspectives, and then constructs the interactive information integration model of virtual reality, and the final output value is the cognitive load value of users, such that the cognitive load can be quantified. As shown in Figure 1, in order to realize the functions in the interactive system, users use visual, auditory, and other cognitive channels to analyze the task, and eye movement is studied to collect the user's eye movement behaviors under single-channel, double-channel, and triple-channel conditions. The user's cognitive load in the virtual reality system can then be quantified.

### 2.2. Construction of Cognitive Load Quality Evaluation Model

The evaluation model is generally composed of three layers: the first layer is the basic layer, that is, the evaluation quality characteristics; the second layer is the middle layer, which is further explanation of the first layer, that is, the characteristics of the mass quantum; and the third layer is the measurement index. Based on the hierarchical partition theory of the quality evaluation model, this paper analyzes the attributes of a virtual reality interactive system, takes the size of user cognitive load as the quality characteristics of a virtual reality interactive system quality evaluation model, deduces the quality sub-characteristics, and finally establishes the cognitive load quality evaluation model of the virtual reality interactive system with the eye movement technical index as the measurement index, as shown in Figure 2.

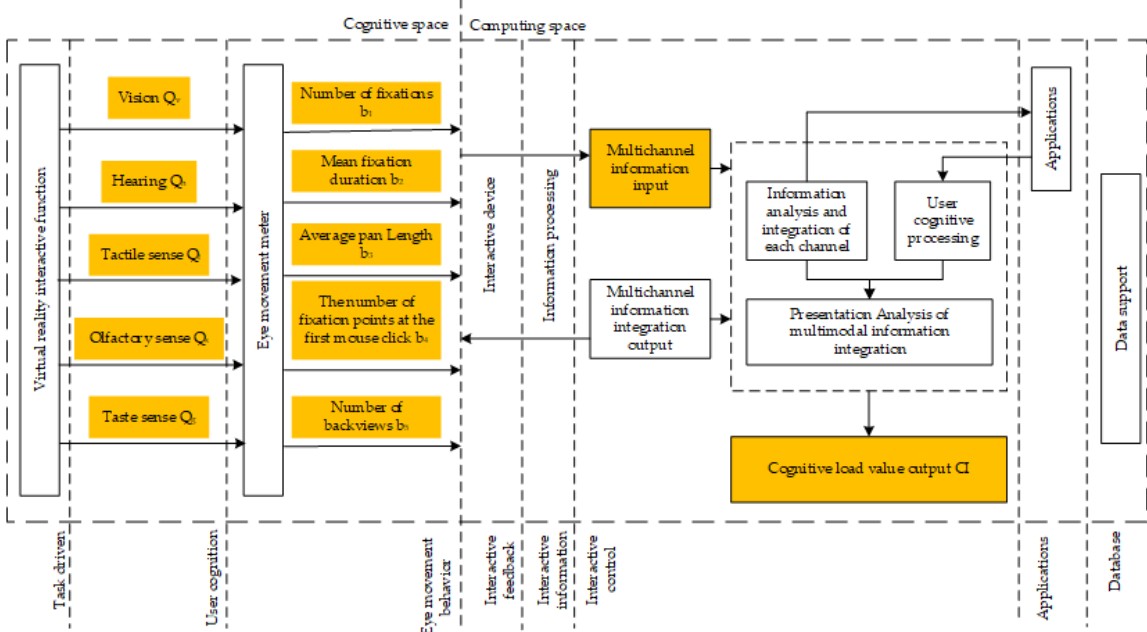

**Figure 1.** Multi-modal interactive information integration model in a virtual reality system.

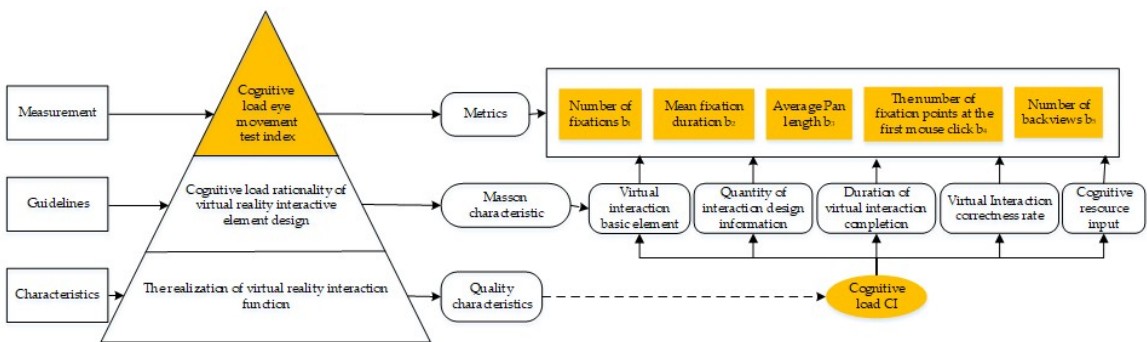

**Figure 2.** Eye movement assessment model of cognitive load in a virtual reality system.

*2.3. Physiological Index of Cognitive Load Based on an Eye Movement Experiment*

In an eye movement experiment as a method of implicitly obtaining cognitive load, the visual behavior recorded by the eye movement instrument is more intuitive than the operating behavior for reflecting the cognitive awareness of users. As the most widely used cognitive load assessment method, eye movement technology is mainly based on the number of fixation points, average fixation duration, average saccade length, number of fixation points at the first mouse click, number of backward-looking times, and other experimental data [26] in order to objectively and scientifically evaluate the cognitive load of a virtual reality interactive system. Therefore, this paper chooses eye movement technology as the experimental approach to establish the cognitive load evaluation model based on the probabilistic neural network.

1. Number of Fixations

The number of fixation points is proportional to the cognitive load of the virtual reality intersection system. The greater the number of fixation points, the larger the cognitive load is, and vice versa [27,28]. Therefore, the number of fixation points is introduced as a physiological index to measure the cognitive load of households.

2. Mean Fixation Duration

The more information you carry, the longer your eyes stay fixed, and the more cognitive load you have. To some extent, this evaluation index can reflect the cognitive load of users intuitively [28–30]. For this reason, the average fixation duration is used as a physiological index to evaluate the cognitive load of users.

3. Average Pan Length

Scanning length is used to calculate the length of the bevel according to the coordinates of the fixation point, which is mainly used to analyze the path [31,32] to be scanned, and thus to analyze the size of the cognitive load of the user.

4. The Number of Fixation Points at the First Mouse Click

Before the first mouse click, the greater the number of the user's fixation points, the higher the user's recognition degree, and the smaller the user's cognitive load [33,34]. This index is inversely proportional to the cognitive load.

5. Number of Back Views

The number of backward-looking views represents the cognitive impairment of the user [35]. The causes of backward-looking include: (1) cognitive bias of the subjects and (2) a big contrast between the cognitive object and the subjects' mental image symbols. Users need to recognize them repeatedly to establish and construct new mental image symbols.

## 3. Methods

### 3.1. Cognitive Load Evaluation Model Based on the Probabilistic Neural Network

**Theorem 1.** *The user's cognitive domain is represented by U, and the cognitive domain is composed of cognitive channels C, expressed as:*

$$U = \begin{bmatrix} C_\alpha \ C_\beta \ C_\lambda \ \cdots \end{bmatrix} \tag{1}$$

*where $C_\alpha, C_\beta, C_\lambda \cdots$ each represent a kind of cognitive channel, and the cognitive behavior set of users under the comprehensive effect of each cognitive channel is represented as B. Then, the set of cognitive behaviors of the user is:*

$$B = [b_1 \ b_2 \ b_3 \ \cdots \ b_s] \tag{2}$$

*where $b_i$ is the index of the user's cognitive behavior, $0 < i < s$.*

Taking the eye movement characteristic parameters in the virtual reality interactive system as the input layer and the cognitive load as the output layer, a cognitive load quantification model is constructed, as shown in Figure 3.

- Input layer: This refers to eye movement data of the entire virtual reality tunnel rescue mission, such as the number of fixation points, in a single vision channel, dual vision-audio channel, dual vision-tactile channel, and three visual-audio-tactile channels. It also includes average gaze duration, average squint length, number of gaze points to the first mouse click, number of gaze times, etc.
- Fusion layer: This refers to incorporating the acquired data into the cognitive load quantification model based on the probabilistic neural network for data collation.
- Output layer: This refers to the value of the final output after the data fusion processing, which is the value of the cognitive load quantified by the tester under a certain conditional channel.

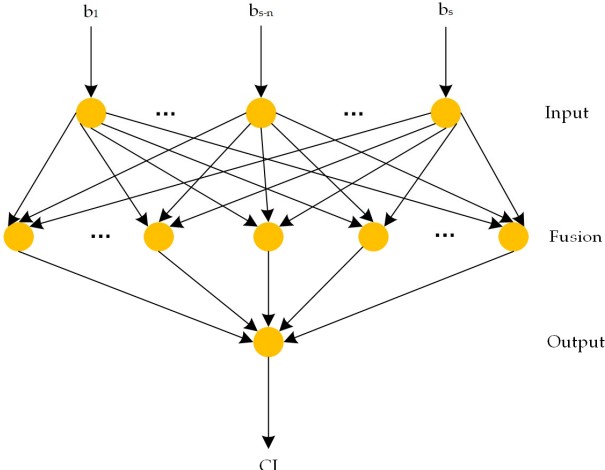

**Figure 3.** Probabilistic neural network model.

There are $y$ scheme values and $s$ eye movement indicators. The matrix of the eye movement indicator data of each scheme is as follows:

$$
B' = \begin{bmatrix}
b_{11}' & b_{12}' & \cdots & b_{1s}' \\
b_{21}' & b_{22}' & \cdots & b_{2s}' \\
\vdots & \vdots & \vdots & \vdots \\
b_{y1}' & b_{y2}' & \cdots & b_{ys}'
\end{bmatrix}
$$

The eye movement index matrix is $B' = \left(b_{ij}'\right)_{y \times s}$. Each column of the matrix represents eye movement indicator data, and each row represents a test value. As the units of each indicator data are different, it is difficult to directly compare the data, so it is necessary to normalize the data of each column, perform linear transformation of the original data, and map the result value to $[0-1]$. If the cognitive load value increases with the increase of each set of indicator data, the transfer function is as follows:

$$
b_{ip} = \frac{b_{ip}' - \min\left\{b_{ip}' | i = 1, 2, \cdots, s\right\}}{\max\left\{b_{ip}' | i = 1, 2, \cdots, s\right\} - \min\left\{b_{ip}' | i = 1, 2, \cdots, s\right\}}
\tag{3}
$$

Conversely, the conversion function is

$$
b_{ip} = \frac{\max\left\{b_{ip}' | i = 1, 2, \cdots, s\right\} - b_{ip}'}{\max\left\{b_{ip}' | i = 1, 2, \cdots, s\right\} - \min\left\{b_{ip}' | i = 1, 2, \cdots, s\right\}}
\tag{4}
$$

where max is the maximum value of the indicator data, min is the minimum value of the indicator data, $p = y * s$, and the improved matrix $B = \left(b_{ij}\right)_{y \times s}$ is:

$$
B = \begin{bmatrix}
b_{11} & b_{12} & \cdots & b_{1s} \\
b_{21} & b_{22} & \cdots & b_{2s} \\
\vdots & \vdots & \vdots & \vdots \\
b_{y1} & b_{y2} & \cdots & b_{ys}
\end{bmatrix}
$$

When $Z_j = \begin{bmatrix} b_{1j} \ b_{2j} \ \cdots \ b_{yj} \end{bmatrix}^T$, $Z$ is the dimensional column vector of $y$. The goal of this paper is to find an estimation function, $\widehat{Z} = Z(b)$, such that the mean square error represented by:

$$error = \sum_{j=1}^{s}\left(Z_j - \widehat{Z}_j\right)^2 \tag{5}$$

is minimized. For a given set of column vectors $B = B_i^T = [b_{i1}\ b_{i2}\ \cdots\ b_{is}]^T$, $Z = Z_i = \left[b_{1j}\ b_{2j}\ \cdots\ b_{yj}\right]^T$. According to the conditional expectation, the estimated function is:

$$\widehat{Z}(B) = \frac{\int_{-\infty}^{\infty} Zf(B,Z)dZ}{\int_{-\infty}^{\infty} f(B,Z)dZ} \tag{6}$$

where $f(B,Z)$ is the joint probability distribution function of $(B,Z)$. The estimate for $f(B,Z)$ is:

$$\widehat{f}(B,Z) = \frac{1}{(2\pi)^{\frac{s+1}{2}}\sigma^{s+1}}\frac{1}{y} \times \sum_{i=1}^{y} \exp\left[-\frac{\left(B - B_i^T\right)^T\left(B - B_i^T\right)}{2\sigma^2}\right]\exp\left[-\frac{(Z - Z_i)^2}{2\sigma^2}\right] \tag{7}$$

where $\sigma$ is the smoothing parameter; $s$ is the dimension of $B$, that is, $s$ kinds of eye movement index parameters are selected; and $y$ is the number of samples, that is, the number of schemes. Then:

$$D_i^2 = \left(B - B_i^T\right)^T\left(B - B_i^T\right) \tag{8}$$

where the physical meaning of $D_i$ is the distance from each input eye movement index to the sample point $i$, which is the Euclidean distance. Here, $\sigma = \frac{\max\{D_i|i=1,2,\cdots,y\}}{\sqrt{y}}$. Substituting $\widehat{f}(B,Z)$ for $f(B,Z)$ in Equation (6), substituting in Equation (8), and exchanging the order of summation and integral number, this can be simplified to obtain:

$$\widehat{Z} = \frac{\sum\limits_{i=1}^{y} Z_i \exp\left(-\frac{D_i^2}{2\sigma^2}\right)}{\sum\limits_{i=1}^{y} \exp\left(-\frac{D_i^2}{2\sigma^2}\right)} \tag{9}$$

$$CI' = E * \widehat{Z} \tag{10}$$

The data is then normalized so that the cognitive load value is in the range of $[0-1]$, and the normalized processing function is as follows:

$$CI_l = \frac{CI_l' - \min\{CI_l'|l = 1,\cdots,p\}}{\max\{CI_l'|l = 1,\cdots,p\} - \min\{CI_l'|l = 1,\cdots,p\}} \tag{11}$$

where $CI$ is the final output, the cognitive load value, of which $E = [1\ 1\ 1\ 1\ 1]$ and $p = y * s$.

### 3.2. Evaluation Index

The experimental output error is defined as:

$$E_k = \frac{1}{2}\left(CI_k^* - CI_k\right)^2 \tag{12}$$

where $k$ denotes the number of cognitive channels, $CI_k^*$ denotes the number of subjective scores for the cognitive load of the virtual reality interactive system under $k$ cognitive channels, and $CI_k$ denotes the value calculated by the user cognitive load evaluation model under $k$ cognitive channels.

In this paper, the maximum absolute error $ER_1$ and the relative mean square error $ER_2$ are used to evaluate the evaluation effect of the model, and the calculation method is as follows:

$$ER_1 = \max_k \left| \frac{CI_k - CI_k^*}{CI_k} \right| \times 100\% \tag{13}$$

$$ER_2 = \sqrt{\frac{1}{H}\sum_{k=1}^{H}\left(\frac{CI_k - CI_k^*}{CI_k}\right)^2} \times 100\% \tag{14}$$

where $H$ is the total number of channel classes.

## 4. Application Instance

### 4.1. Experimental Design

A VR tunnel emergency rescue system mainly obtains rescue information using a visual reading; the auditory system acquires tunnel rescue information, such as tunnel wind sound, water drops sounds, etc., and obtains rescue information; and the touch sense is initiated by touching the handle to obtain the selected rescue information. This paper is mainly focused on the virtual reality system. The tester wore virtual reality equipment and eye-moving equipment; completed the selection of vehicles by visual, auditory, and tactile systems; selected rescue teams; detected life; opened life channels; and provided rescue channels and other rescues. Based on the VR tunnel emergency rescue system, the main focus was on vision. If the experiment was not completed without the visual channel, this paper only studied the cognitive load under the visual $Q_v$, visual-auditory $Q_v - Q_h$, visual-tactile $Q_v - Q_t$, and visual-auditory-tactile $Q_v - Q_h - Q_t$ channels. The experimental task was carried out in the Key Laboratory of Modern Manufacturing Technology of the Ministry of Education of Guizhou University, China, to keep the environment quiet and the light stable, eliminating all interference experimental factors. The study included a task with four layers of cognitive load, from a single channel to three channels. Specifically, the four tasks were as follows:

- Visual channel: The sound equipment and handle of the emergency rescue system of the VR tunnel were switched off, and the tester obtained the rescue mission information only through the visual channel to complete the rescue mission.
- Visual-auditory: The handle of the VR tunnel emergency rescue system was turned off, and the tester obtained rescue mission information through visual and auditory functions to complete the rescue mission.
- Visual-tactile: The sound equipment of the VR tunnel emergency rescue system was turned off. The tester obtained rescue mission information through visual and tactile sensation and completes the rescue mission.
- Visual-auditor-tactile: The tester obtained the rescue information through visual reading; the auditory system acquires the tunnel rescue information, such as the tunnel wind sound, the water drops sounds, etc., and obtains the rescue information; the handle was touched to obtain the selected rescue information to complete the rescue task.

For each tester, random numbering was performed, and each tester had a preparation time of 1 min. The tester's task schedule is shown in Table 1. The experimenter completed the tunnel emergency rescue task through virtual reality equipment, and acquired the eye movement data in the process of completing the task by using the strap-back eye tracker of Xintuo Inki Technology Company. For example, the number of fixations, mean fixation duration, average pan length, number of fixation points at the first mouse click, and number of back-views were obtained. Subjective measurement and self-assessment is widely used as a measure of cognitive load [9,36–38], which can detect small changes in cognitive load with a relatively good sensitivity [39]. Therefore, at the end of the experiment, in order to verify the usability of the cognitive load evaluation model based on the probabilistic neural

network and reduce the subjective measurement error of cognitive load, all the subjects were required to complete the cognitive load questionnaire immediately after completing the task.

**Table 1.** Testers distribution table.

| Cognitive Channel | | Subject Serial Number |
|---|---|---|
| Single-channel $k = 1$ | Vision $Q_v$ | 1, 2, 3, 4, 5 |
| Dual-channel $k = 2$ | Visual-auditory $Q_v - Q_h$<br>Visual-tactile $Q_v - Q_t$ | 6, 7, 8, 9, 10<br>11, 12, 13, 14, 15 |
| Three channels $k = 3$ | Visual-auditory-tactile<br>$Q_v - Q_h - Q_t$ | 16, 17, 18, 19, 20 |

### *4.2. Select Subjects*

Twenty virtual reality game lovers from Guizhou University were selected as subjects, aged between 24 and 30. The subjects were in good health, had no bad habits (smoking, drinking, etc.), were colorless, were weak or color blind, and their eyesight or corrected eyesight was 1.0. Before the experiment, it was confirmed that the participants did not drink alcohol or coffee or other stimulant drinks on the day of the experiment, and they signed the agreement voluntarily under the condition that they were familiar with the "informed consent form."

### *4.3. Experimental Device*

In the experiment of the Key Laboratory of Modern Manufacturing Technology of Guizhou University, a 29-inch LED screen and a resolution Lenovo computer were used. The emergency rescue mission of the tunnel was completed using China's HTC VIVE virtual reality device, and eye movement data was acquired through the new Tony Inge's EyeSo Ee60 telemetry eye tracker.

### *4.4. Experimental Variables*

#### 4.4.1. Independent Variable

As shown in Table 2, the cognitive channel was an independent variable, and the participants completed the emergency rescue task of the VR tunnel with different cognitive channels.

**Table 2.** Independent variable.

| Number of Cognitive Channels | Classes | |
|---|---|---|
| Single-channel $k = 1$ | Vision $Q_v$ | |
| Dual-channel $k = 2$ | Visual-auditory $Q_v - Q_h$ | Visual-tactile $Q_v - Q_t$ |
| Three channels $k = 3$ | Visual-auditory-tactile $Q_v - Q_h - Q_t$ | |

#### 4.4.2. Dependent Variable

In order to verify the rationality of the cognitive load evaluation model and analyze the subjective scores of the cognitive load of different subjects, the cognitive load scores were $[0 - 1]$, with 0 for a low subjective load and 1 for a high subjective load, as shown in Table 3. The result is a subjective evaluation of the cognitive load of the virtual reality interactive system. The participants' questionnaire is shown in Table 4.

**Table 3.** Cognitive load rating.

| Cognitive Load Layer | 0 | 0.2 | 0.4 | 0.6 | 0.8 | 1 | 0.1, 0.3, 0.5, 0.7, 0.9 |
|---|---|---|---|---|---|---|---|
| Meaning | Extremely low cognitive load | Cognitive load is intensely low | Cognitive load was significantly lower | Cognitive load was significantly high | Cognitive load is intensely high | Extremely high cognitive load | The intermediate value of the neighboring judgment |

**Table 4.** Subjective cognitive load questionnaire.

| Cognitive Channel | | Cognitive Load |
|---|---|---|
| Single-channel $k = 1$ | Vision $Q_v$ | 0.7 |
| Dual-channel $k = 2$ | Visual-auditory $Q_v - Q_h$ | 0.5 |
| | Visual-tactile $Q_v - Q_t$ | 0.2 |
| Three channels $k = 3$ | Visual-auditory-tactile $Q_v - Q_h - Q_t$ | 0.1 |

As the number of cognitive channels changed, so does the eye movement index data, as shown in Table 5.

**Table 5.** Dependent variable.

| Cognitive Channel | | Eye Movement Index | | | | |
|---|---|---|---|---|---|---|
| | | Eye Movement Index $b_1$ | Mean Fixation Duration $b_2$ | Average Pan Length $b_3$ | The Number of Fixation Points at the First Mouse Click $b_4$ | Number of Back Views $b_5$ |
| Single-channel $k = 1$ | Vision $Q_v$ | 0.2812 | 0.7555 | 0.9492 | 0.5556 | 0.6000 |
| Dual-channel $k = 2$ | Visual-auditory $Q_v - Q_h$ | 0.3438 | 0.6823 | 0.5024 | 0.3333 | 0.5000 |
| | Visual-tactile $Q_v - Q_t$ | 0.2812 | 0.5115 | 0.6378 | 0.0000 | 0.4000 |
| Three channels $k = 3$ | Visual-auditory-tactile $Q_v - Q_h - Q_t$ | 0.2500 | 0.2537 | 0.0030 | 0.0000 | 0.2000 |

### 4.4.3. Control Disturbance Variable

In order to avoid repeated experiments and to remember the influence of the VR tunnel emergency rescue system environment and task on the cognitive load supervisor score, each participant could only complete one kind of modal cognitive experiment, such as the one-way to visual cognitive experiment, which was arranged as shown in Table 1.

### 4.5. Experimental Results

The cognitive load of the emergency rescue system in the VR tunnel in different cognitive channel environments was objectively evaluated. The results are shown in Table 6.

**Table 6.** Subjective cognitive load.

| Cognitive Channel Category | Single Channel $k=1$ | Dual Channel $k=2$ | | Three Channel $k=3$ |
|---|---|---|---|---|
| | Vision $Q_v$ | Visual-Auditory $Q_v-Q_h$ | Visual-Tactile $Q_v-Q_t$ | Visual-Auditory-Tactile $Q_v-Q_h-Q_t$ |
| Cognitive load | 0.7 | 0.5 | 0.2 | 0.1 |
| | 0.6 | 0.5 | 0.4 | 0.2 |
| | 0.9 | 0.7 | 0.5 | 0.05 |
| | 0.7 | 0.4 | 0.3 | 0 |
| | 0.7 | 0.4 | 0.4 | 0.06 |

Table 7 shows the data of eye movement indices during the emergency rescue of the VR tunnel under different cognitive channels, which have been normalized.

**Table 7.** Normalized eye movement index data.

| Cognitive Channel | | Eye Movement Index | | | | | |
|---|---|---|---|---|---|---|---|
| | | Eye Movement Index $b_1$ | Mean Fixation Duration $b_2$ | Mean Fixation Duration $b_3$ | The Number of Fixation Points at the First Mouse Click $b_4$ | Number of Back Views $b_5$ | Cognitive Load |
| Single-channel $k = 1$ | Vision $Q_v$ | 0.2812 | 0.7555 | 0.9492 | 0.5556 | 0.6000 | 0.6788 |
| | | 0.8438 | 0.4468 | 0.7731 | 1.0000 | 0.5000 | 0.5962 |
| | | 1.0000 | 0.6051 | 1.0000 | 0.6667 | 0.6000 | 1.0000 |
| | | 0.6250 | 0.5862 | 0.9176 | 0.5556 | 1.0000 | 0.7241 |
| | | 0.5000 | 0.5016 | 0.9902 | 0.6667 | 0.8000 | 0.7370 |
| Dual-channel $k = 2$ | Visual-auditory $Q_v - Q_h$ | 0.3438 | 0.6823 | 0.5024 | 0.3333 | 0.5000 | 0.5043 |
| | | 0.6250 | 0.5288 | 0.6810 | 0.4444 | 0.3000 | 0.4934 |
| | | 0.3750 | 1.0000 | 0.6631 | 0.3333 | 0.3000 | 0.6684 |
| | | 0.4375 | 0.2424 | 0.8045 | 0.1111 | 0.4000 | 0.3620 |
| | | 0.7188 | 0.0000 | 0.5541 | 0.6667 | 0.7000 | 0.4344 |
| | | 0.2812 | 0.5115 | 0.6378 | 0.0000 | 0.4000 | 0.2279 |
| | Visual-tactile $Q_v - Q_t$ | 0.3125 | 0.2702 | 0.4161 | 0.3333 | 0.3000 | 0.4133 |
| | | 0.0938 | 0.5154 | 0.3611 | 0.3333 | 0.3000 | 0.4516 |
| | | 0.5625 | 0.4137 | 0.3623 | 0.1111 | 0.4000 | 0.2586 |
| | | 0.0625 | 0.3982 | 0.4966 | 0.6667 | 0.5000 | 0.3646 |
| Three channels $k = 3$ | Visual-auditory-tactile $Q_v - Q_h - Q_t$ | 0.2500 | 0.2537 | 0.0030 | 0.0000 | 0.2000 | 0.0751 |
| | | 0.0312 | 0.0051 | 0.0000 | 0.1111 | 0.2000 | 0.2011 |
| | | 0.1250 | 0.0935 | 0.2113 | 0.0000 | 0.0000 | 0.0523 |
| | | 0.0000 | 0.2675 | 0.1987 | 0.1111 | 0.1000 | 0.0000 |
| | | 0.2500 | 0.5775 | 0.1449 | 0.1111 | 0.1000 | 0.0621 |

## 5. Discussion

### 5.1. Correlation Analysis of Eye Movement Parameters and Cognitive Load of Users

Users' cognitive load obtained from a single type of eye movement data was limited and one-sided, which cannot accurately reflect the needs of users' interests. Therefore, it is necessary to integrate the data and establish a model of users' cognitive load based on an eye movement experiment. Additionally, it is necessary to analyze the correlation between eye movement data and the cognitive load.

In this paper, the Pearson correlation test was used to test the relationship between eye movement parameters and cognitive load, so as to improve the theoretical premise of the cognitive load evaluation. The results of the correlation analysis were obtained and can be viewed in Table 8.

**Table 8.** Correlation between each eye movement characteristic parameter and cognitive load.

| Eye Movement Characteristic Parameter | Eye Movement Index $b_1$ | Mean Fixation Duration $b_2$ | Mean Fixation Duration $b_3$ | The Number of Fixation Points at the First Mouse Click $b_4$ | Number of Back Views $b_5$ |
|---|---|---|---|---|---|
| $r$ | 0.679878252 | 0.559834694 | 0.863182783 | 0.754462615 | 0.754440400 |

As can be seen from Table 8, the characteristic parameters of each eye movement index were significantly correlated with the cognitive load of users to varying degrees, and the high correlation between the eye movement index and the cognitive load is demonstrated once again. Average saccade length was more highly correlated with cognitive load than other parameters.

### 5.2. Model Output Analysis

Comparative analysis of the cognitive load evaluated by the probabilistic neural network model and actual cognitive load is shown in Figure 4, and the fitting degree is high. From Figure 4, it can

be seen that the cognitive load evaluation model is close to the actual result, which indicates that the evaluation effect of this model is better.

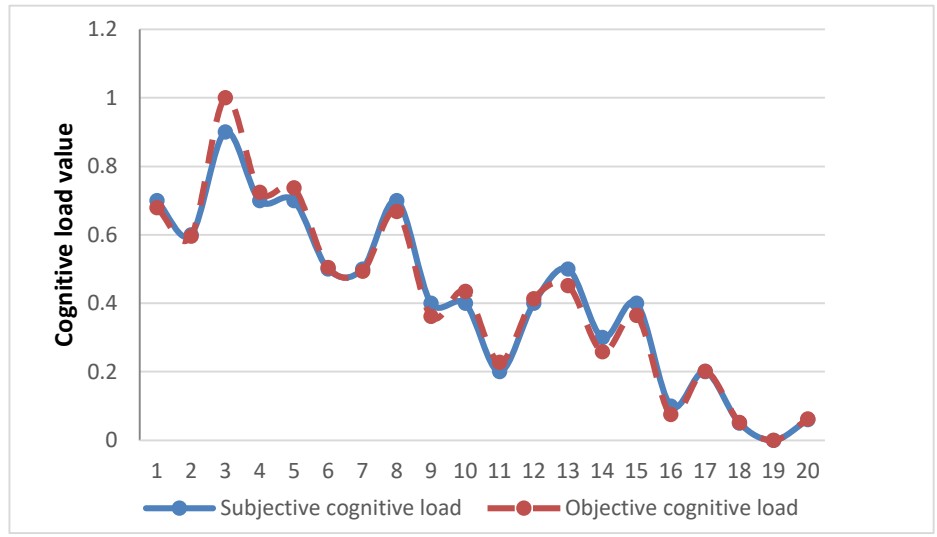

**Figure 4.** The cognitive load evaluated by the probabilistic neural network model is compared with the actual cognitive load.

At the same time, in order to understand the accuracy of the model used, the maximum absolute error and relative mean square error were used to evaluate the model, and the evaluation results are shown in Table 9.

**Table 9.** Maximum absolute error and relative mean square error.

| Cognitive Channel | Subjective Cognitive Load | Cognitive Load Quantified Value | Absolute Error | Average Absolute Error | Mean of Mean Absolute Error | Relative Mean Square Error | Mean Relative Mean Square Error |
|---|---|---|---|---|---|---|---|
| Vision | 0.6788 | 0.7 | 0.0312 | 0.1 | | 0.0989 | |
| | 0.5962 | 0.6 | 0.0064 | | | | |
| | 1 | 0.9 | 0.1 | | | | |
| | 0.7241 | 0.7 | 0.0333 | | | | |
| | 0.737 | 0.7 | 0.0502 | | | | |
| Visual-auditory | 0.5043 | 0.5 | 0.0085 | 0.105 | | 0.1133 | |
| | 0.4934 | 0.5 | 0.0134 | | | | |
| | 0.6684 | 0.7 | 0.0473 | | | | |
| | 0.362 | 0.4 | 0.105 | | | | |
| | 0.4344 | 0.4 | 0.0792 | | 0.107575 | | 0.127675 |
| Visual-tactile | 0.2279 | 0.2 | 0.1224 | 0.1601 | | 0.2321 | |
| | 0.4133 | 0.4 | 0.0322 | | | | |
| | 0.4516 | 0.5 | 0.1072 | | | | |
| | 0.2586 | 0.3 | 0.1601 | | | | |
| | 0.3646 | 0.4 | 0.0971 | | | | |
| Visual-auditory-tactile | 0.0751 | 0.08 | 0.0652 | 0.0652 | | 0.0664 | |
| | 0.2011 | 0.2 | 0.0055 | | | | |
| | 0.0523 | 0.05 | 0.044 | | | | |
| | 0 | 0 | 0 | | | | |
| | 0.0621 | 0.06 | 0.0338 | | | | |

In general, the mean absolute error was 10.7575% and the mean relative mean square error was 12.7675%. At the same time, it can be seen from the cognitive load evaluation results of each cognitive channel that the maximum absolute error was 16.01%, the minimum absolute error was 6.52%, the maximum relative mean square error was 23.21%, and the minimum relative mean square error was 6.64%. This shows that the cognitive load evaluation model based on the probabilistic neural network had a high precision, and the cognitive load model proposed in this paper had a

good reliability and can accurately evaluate the cognitive load value of users under different cognitive channels, so as to effectively improve the design rate of the virtual reality interaction system and the user experience.

## 6. Conclusions

In this paper, the eye movement behavior of the experimenters in a virtual reality interactive environment was studied, and the cognitive load was calculated using the eye movement index such that the cognitive load could be quantified. Eye movement data were recorded using an eye movement instrument, and the subjective cognitive load of the current interactive system was investigated using a questionnaire. The conclusions are as follows.

Based on the experimenter's eye movement experiment, the number of fixation points, the average fixation duration, the average saccade length, the number of fixation points clicked during the first time, the number of backward-looking views, and other eye movement data were extracted, the user's cognitive load quantification model in the virtual reality interactive system was constructed by combining the probabilistic neural network.

From the results of the study, it can be seen that there was a significant correlation between each eye movement characteristic parameter and the cognitive load, which indicates that the eye movement index can directly reflect the cognitive load under the interaction of users, thus providing a basis for the study of cognitive load quantification.

The results show that the absolute error of the user cognitive load based on the probabilistic neural network and the subjective cognitive load value of the tester was 6.52%–16.01%, and the relative mean square error is 6.64%–23.21%, indicating that the method has a high precision.

**Author Contributions:** Conceptualization, X.X. and J.L.; methodology, X.X.; validation, X.X., J.L., and N.D.; formal analysis, X.X.; investigation, X.X.; resources, X.X.; data curation, X.X; writing—original draft preparation, X.X.; writing—review and editing, X.X.; visualization, N.D.; supervision, J.L.; project administration, J.L.

**Funding:** This research was supported by the Natural Science Foundation of China (Nos. 51865004, 2014BAH05F01) and the Provincial Project Foundation of Guizhou, China (Nos. [2018]1049, [2016]7467).

**Acknowledgments:** The authors would like to convey their heartfelt gratefulness to the reviewers and the editor for the valuable suggestions and important comments which greatly helped them to improve the presentation of this manuscript.

**Conflicts of Interest:** The authors declare no conflict of interest.

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
