# Peer review of "Research on the Quantitative Method of Cognitive Loading in a Virtual Reality System"

_information, doi:10.3390/info10050170_

Round 1

Reviewer 1 Report

This article discusses a predictive cognitive load model (a probabilistic neural network model) for virtual reality (VR) tasks of the "tunnel rescue" type. The model exploits eye movement parameters such as number of fixations, average duration of fixations, average saccade length, number of backward looks and other of experienced VR game players while performing the experimental task. The measured eye movement parameters are fed into a neural network architecture with three layers (input layer, fusion layer, output layer). The output layer gives the fused probabilistic estimate of cognitive load. It is stated that the model can be tested under three functional regimes (hypotheses): 1) a single visual channel regime 2) a combined visual-tactile channel regime and 3) a combined visual-auditory channel regime. The subjectively experienced cognitive task load was assessed by having the subjects fill in a questionnaire. The results of the model simulations show that the probabilistic predictions of the neural network model fit the subjectively experienced cognitive load data with great precision. 

To warrant publication, the article needs to be revised (major revisions are required), with attention to the following:

1) The experimental task (virtual tunnel rescue task) is not described in sufficient detail. To have some idea of the objective task difficulty inducing significant variations in cognitive load across users, the reader needs to know what had to be done exactly, in how many steps, how long it takes an expert to perform each of these steps, and with which (expected) rate of task success. The individual data of each subject need to be shown to see how easy or difficult it was for each of them to perform the given task in light of the expected performance of an expert user. The subjectively experienced cognitive load of the subjects (results of the questionnaire) only makes sense when presented in conjunction with objective measures of task difficulty. 

2) This point is directly related to point 1). In the text, the authors refer to the results from the questionnaire in terms of "objective cognitive load" data, which is not only misleading but incorrect. Data from a questionnaire give an idea of "subjectively experienced cognitive load". Please revise accordingly.

3) Since eye movement parameters only were recorded, the single "visual channel" regime hypothesis only is validated experimentally. It remains unclear how parameters relative to touch and sound were measured in the task, and taken into account for the combined visual-tactile and visual-auditory channel assumptions supposedly generating model predictions. 

4) The functional properties of each layer of the probabilistic neural network model (Figure 3) deserve to be described more clearly, especially in the light of point 3).

5) The legend of Figure 4 needs to be changed. So far, we have no information about the "objective cognitive load" (see point 2)), only the subjective data from the questionnaire. The x and the y axes of the Figure need clear legends.

After all major revisions have been performed on the contents of the paper, I strongly recommend having the text checked carefully by a native English speaker (colleague or editor).

Author Response

Dear review editor:

First of all, thank you for your valuable suggestions for our paper! We made these changes one by one in the first place. The following is the explanation attached with the paper. Please review it. Wish you a good job and a happy life!

Point 1: The experimental task (virtual tunnel rescue task) is not described in sufficient detail. To have some idea of the objective task difficulty inducing significant variations in cognitive load across users, the reader needs to know what had to be done exactly, in how many steps, how long it takes an expert to perform each of these steps, and with which (expected) rate of task success. The individual data of each subject need to be shown to see how easy or difficult it was for each of them to perform the given task in light of the expected performance of an expert user. The subjectively experienced cognitive load of the subjects (results of the questionnaire) only makes sense when presented in conjunction with objective measures of task difficulty. 

Response 1: The revision here relates to the 4.1 Experiment Design section. We have added the section of Experiment Design, which describes the task in detail and the participants and the experiment in a broader way. See the original text.

Point 2: This point is directly related to point 1). In the text, the authors refer to the results from the questionnaire in terms of "objective cognitive load" data, which is not only misleading but incorrect. Data from a questionnaire give an idea of "subjectively experienced cognitive load". Please revise accordingly.

Response 2: The revisions here relate to abstracts, 4.1 experimental designs, and 6 conclusions. Amend the summary section to read as follows: The model was validated by using eye movement data and Subjective cognitive load data.” 6 amend that conclusion to read as follow: Eye movement data were recorded by eye movement instrument, and Subjective cognitive load of current interactive system was investigated by questionnaire.” 4.1 Refer to the original text for the revision of the experimental design.

Point 3: Since eye movement parameters only were recorded, the single "visual channel" regime hypothesis only is validated experimentally. It remains unclear how parameters relative to touch and sound were measured in the task, and taken into account for the combined visual-tactile and visual-auditory channel assumptions supposedly generating model predictions.

Response 3: In this paper, we consider the cognitive load of the subjects in the single visual channel, double visual-auditory channel, double visual-tactile channel, tri-visual-auditory-tactile channel to complete the VR tunnel rescue task. This VR tunnel rescue system is mainly through the visual channel to observe the rescue picture information through hearing the tunnel wind water drops and other rescue information as well as through tactile channel handle to touch the object sent out by the vibration information to complete the rescue task. Eye movement parameters are only a physiological measure of cognitive load. See 4.1 Experiment section for details.

Point 4: The functional properties of each layer of the probabilistic neural network model (Figure 3) deserve to be described more clearly, especially in the light of point 3).

Response 4: The modification herein relates to the 3.1 Probabilistic Neural Network Based Cognitive Load Quantification Model section. Input layer: It refers to eye movement data of the entire virtual reality tunnel rescue mission, such as the number of fixation points, in a single vision channel, dual vision-audio channel, dual vision-tactile channel, and three visual-audio-tactile channels. Average gaze duration, average squint length, number of gaze points to the first mouse click, number of gaze times, etc. Fusion layer: It refers to bringing these acquired data into the cognitive load quantification model based on probabilistic neural network for data collation. Output layer: It refers to the value of the final output after the data fusion processing is the value of the cognitive load quantified by the tester under a certain conditional channel.

Point 5: The legend of Figure 4 needs to be changed. So far, we have no information about the "objective cognitive load" (see point 2)), only the subjective data from the questionnaire. The x and the y axes of the Figure need clear legends.

Response 5: The Y-axis of FIG. 4 is a cognitive load value, and that X-axis is a subjective cognitive load and a cognitive load quantification value. See the original figure 4

Reviewer 2 Report

This paper presents a procedure for measuring the load cognitive for users of virtual reality, and it was validated some metrics such as eye movement index, fixation duration, number of back views and fixation point at the first mouse click. The stimulus individually and together analyzed were the vision, hearing, and tactile.  In general, the paper was well driven, and some minor corrections should be carried out.

Line 26: define IVIS.

Introduction: maybe the word "scholar" can be changed by "researcher."

Line 44-47: maybe, these lines are a little wordiness

Section 2.2: use the same word "level" or "layer," not both. It can be confusing.

Figure 1. This figure is a little confusing, check the arrow directions and wide the explanation.

From line 140-289: check the notation and writing of the numbers and equations. e.g., throughout these lines, some numbers are writing as super-index. Besides, I suggest changing the word "formula" by "equation."

Line 167: "A" is defined, but it is not applied. Check equation.

I suggest to broad the description of the participants and experiments, in particular, the VR environments used.

I suggest to wide the explanation of subjective cognitive load questionnaire. Besides, it can be shown in a better way, for reducing space a improve the description. The last column cognitive load (is empty) maybe can add information about how carried out these questions.

Table 4. maybe it can be eliminated. The same information is in table 7.

I suggest to broad the explanation of Figure 4 considering the parameters for building the probabilistic neural network. Besides, this study should be compared with the results of other works.

Line 274 section Conclusions: I consider that item 1. is not a conclusion. I suggest to eliminate the enumeration and to adjust the section. Include future work.

Table 9.  has two columns with the same names (Mean absolute error).

Author Response

Response to Reviewer 2 Comments

Dear review editor:

First of all, thank you for your valuable suggestions for our paper! We made these changes one by one in the first place. The following is the explanation attached with the paper. Please review it. Wish you a good job and a happy life!

Point 1: Line 26: define IVIS.

Response 1: Line 26: In IVIS (In-vehicle information system). Refer to the original text for details.

Point 2: Introduction: maybe the word "scholar" can be changed by "researcher".

Response 2: We have replaced "scholar" with "researcher", as detailed in the introduction

Point 3: Line 44-47: maybe, these lines are a little wordiness.

Response 3: Line 44-46: We've changed it to These studies have focused on the use of physiological methods to assess the cognitive load of users, but have not yet resolved how to construct a quantitative relationship between physiological indicators and cognitive load. "

Point 4: Section 2.2: use the same word "level" or "layer," not both. It can be confusing.

Response 4: Section 2.2: We have agreed to use the word "layer".

Point 5: Figure 1. This figure is a little confusing, check the arrow directions and wide the explanation.

Response 5: Figure 1 has been modified as shown in the original text.

Point 6: From line 140-289: check the notation and writing of the numbers and equations. e.g., throughout these lines, some numbers are writing as super-index. Besides, I suggest changing the word "formula" by "equation."

Response 6: We have used "equality" instead of "formula". See the original text for other modifications.

Point 7: Line 167: "A" is defined, but it is not applied. Check equation.

Response 7: The amendment here relates to section 3.2 and has been amended to read "Among them, k denotes the number of cognizant channels".

Point 8: I suggest to broad the description of the participants and experiments, in particular, the VR environments used.

Response 8: The revision here relates to the 4.1 Experimental Design section, which adds experimental designs to describe the experimenter and the virtual environment, as detailed in the original text.

Point 9: I suggest to wide the explanation of subjective cognitive load questionnaire. Besides, it can be shown in a better way, for reducing space a improve the description. The last column cognitive load (is empty) maybe can add information about how carried out these questions.

Response 9: The revision here relates to the 4.1 Experimental Design section, as detailed in the original text.

Point 10: Table 4. maybe it can be eliminated. The same information is in table 7.

Response 10: First of all, thanks to the reviewer's comments, Table 4 has been adjusted to Table 1. Its main content is to introduce the experimenter's experimental task arrangement, to avoid interference with the experimental results because of the experimenter's own reasons. Table 7 mainly describes the normalized eye movement data of all subjects after completing the test task.

Point 11: I suggest to broad the explanation of Figure 4 considering the parameters for building the probabilistic neural network. Besides, this study should be compared with the results of other works.

Response 11: First of all, I would like to thank the peer reviewer for his comments. This revision involves 5.2 Figure 4, as shown in the original text. The results of this study are compared with those of other work, such as EEG verification, which is the next research direction of this paper. Thank you for your guidance.

Point 12: Line 274 section Conclusions: I consider that item 1. is not a conclusion. I suggest to eliminate the enumeration and to adjust the section. Include future work.

Response 12: This revision relates to the 6 conclusions, as detailed in the original text

Point 13: Table 9.  has two columns with the same names (Mean absolute error).

Response 13: This revision relates to table 9 and is detailed in the original text.

Round 2

Reviewer 1 Report

The authors have substantially revised this paper and provided a clear description of the experimental task, which considerably improves the manuscript and is essential for understanding their model approach and the data. Also, it is now made clear that the neural network model output is compared to subjective cognitive load ratings of the subjects, not to some supposedly objective measure. This clarifies many of the points addressed. The paper still suffers from absence of an objective measure of task diffculty in the different conditions in terms of relative task time performance, but this may not be so critical any more taking into account that the model and the different sensory modality paramenters mirrored by the eye movement data are now more clearly presented. The paper still contains quite a few typos and language errors, which need to be corrected in a careful revision. I strongly recommend using the journal's editing service.

Author Response

Thank you for your suggestions on our paper. We have already revised the paper. See the original text.

Information EISSN 2078-2489 Published by MDPI AG, Basel, Switzerland RSS E-Mail Table of Contents Alert
Back to Top